# Application Prospect and Preliminary Exploration of GelMA in Corneal Stroma Regeneration

**DOI:** 10.3390/polym14194227

**Published:** 2022-10-09

**Authors:** Guanyu Su, Guigang Li, Wei Wang, Lingjuan Xu

**Affiliations:** Department of Ophthalmology, Tongji Hospital, Tongji Medical College, Huazhong University of Science and Technology, Wuhan 430030, China

**Keywords:** GelMA, corneal stroma, regeneration

## Abstract

Corneal regeneration has become a prominent study area in recent decades. Because the corneal stroma contributes about 90% of the corneal thickness in the corneal structure, corneal stromal regeneration is critical for the treatment of cornea disease. Numerous materials, including deacetylated chitosan, hydrophilic gel, collagen, gelatin methacrylate (GelMA), serine protein, glycerol sebacate, and decellularized extracellular matrix, have been explored for keratocytes regeneration. GelMA is one of the most prominent materials, which is becoming more and more popular because of its outstanding three-dimensional scaffold structure, strong mechanics, good optical transmittance, and biocompatibility. This review discussed recent research on corneal stroma regeneration materials and related GelMA.

## 1. Introduction

Keratoconus is a serious blinding eye disease, with an incidence of 0.05% to 0.23% [1]. This incidence range may be attributed to the increased sensitivity of modern diagnostic devices, regional differences with respect to accessibility of health care, and/or differences in study design [2]. However, it is believed that the incidence of keratoconus may increase further as diagnostic techniques improves. Keratoconus is also the most frequent indication for corneal graft surgery at an age younger than 40 years in many countries of the world, so it is considered to be the leading cause of corneal blindness in adolescents [1,3,4]. Typically, the central or paraccentral cornea is progressively thinner and convex in a conical shape forward, leading to high myopia and irregular astigmatism. Patients’ visual acuity is seriously impaired, and they eventually become blind, which seriously affects their employment and quality of life and causes a heavy family and social burden [1,5].

Although the etiology of keratoconus is not fully understood, with several different pathways, its main pathological features are reduced keratocyte densities and collagen fibers [3]. The only treatment available today that tries to strengthen the cornea and halt the progression of keratoconus is corneal crosslinking. However, because the thickness of the corneal stroma does not improve, it does not reverse the lesions that have already occurred [6].Therefore, stromal replacement is the main treatment for severe keratoconus [7]. Stromal replacement can “restore” the normal thickness of the cornea by partially or completely replacing the thinning corneal stroma, but has some limitations. The implantation of a donor cornea into a recipient eye induces scarring at the donor–recipient tissue interfaces [1,7], changes in refraction due to the suboptimal curvature of the implanted tissue [7], and the chronic decompensation of the corneal graft because of the progressive loss of corneal endothelial cells [8] and carries the potential for immunologic rejection due to the foreign cells contained within the donor tissue [8,9]. Although these potential drawbacks of the use of human donor stromal tissue can be significant, the main drawback of stromal replacement with human donor tissue is the global shortage of donated corneas, especially in China [10,11]. According to the literature, for every 70 patients in the world who need a corneal transplant, only one is able to obtain a corneal donor [12]. The implantation of an allogenic corneal lenticule obtained by small-incision lenticule extraction (SMILE) can improve the visual acuity of patients by increasing the thickness of the corneal stroma, but it is also dependent on corneal donors and has great technical difficulties [13,14]. Furthermore, the lenticules produced by SMILE are frequently not thick enough for treatment. Additionally, it is unknown if several lenticules may be implanted at once; whether this approach is physically possible; and whether the recipient cornea will acquire the necessary transparency, strength, and proper shape after receiving multiple lenticules. Femtosecond laser-assisted minimally invasive lamellar keratoplasty can treat advanced keratoconus, but it also requires corneal donors [15]. Therefore, if the corneal stroma thickness can be increased by in situ regeneration while preserving the patient’s own corneal stroma, it will undoubtedly be an ideal treatment.

## 2. Current Status of Polymers Used in Corneal Stromal Regeneration

Similar to epithelial and endothelial regeneration, it would be ideal if the stroma could be regenerated, assisted by an implant, scaffold, cells, or other factors. The hottest researched and widely utilized form of therapy in regenerative medicine is stem cell therapy, which has been used to treat hematological cancers, urinary disorders, periodontal diseases, bone and joint problems, and ocular surface conditions [16,17,18]. Ocular or extra-ocular mesenchymal stem cells (MSCs) can differentiate into adult keratocytes both in vitro and in vivo, but corneal-stroma-derived MSCs (such as corneal stromal stem cells) have greater differentiation potential than other sources of MSCs [19,20]. A door has been opened for corneal stromal regeneration by stem cell treatment. However, simple injection of stem cells into the corneal stroma has a low rate of differentiation, requires a lot of time, and only slightly thickens the stroma, defeating the objective of an effective treatment [7,19,21]. Studies have found that the use of various biomaterial scaffolds to deliver stem cells has become a more effective regenerative medicine strategy [22,23], providing a consult for the regeneration of the corneal stroma.

An acellular porcine corneal stromal scaffold solves the problem of the donor shortage, but it is difficult for recipient keratocytes to grow into it due to dense tissue. The scales of the fish can be decellularized and decalcified to produce a scaffold compatible with the culture of corneal cells [24,25]. The fish-scale-derived scaffold was successfully used as an emergency patch graft to temporarily seal the cornea and maintain the integrity of the anterior chamber and eye globe for up to four days [26]. The implantation of a cell-free recombinant human collagen scaffold into the corneal stroma resulted in the regenerated neo-corneas being stably integrated without rejection, without the long immunosuppression regime needed by patients after 4 years [27]. However, host keratocytes were still not fully integrated into the scaffold [27]. Some polymer scaffolds and electrospinning scaffolds can be loaded with stem cells, but the transparency is not enough. Some researchers have used synthetic polymers as a substrate for engineering corneal stroma because they have adjustable mechanical properties; some of these scaffolds have the capacity for inducing the differentiation of human stromal stem cells into keratinocyte lineage, and there were some weaknesses in the optical properties of the scaffold [23]. Moreover, most scaffolds are still implanted as lamellar grafts, which are highly traumatic to the cornea and cause scar astigmatism after implantation. Injectable hydrogels can provide a scaffold for in situ tissue regrowth and regeneration, yet gel degradation before tissue reformation limits the gels’ ability to provide physical support [28,29]. It can be concluded that a common scaffold is difficult to fuse with keratocytes or degrades too fast, which is the bottleneck of corneal stromal regeneration. The ideal scaffolds must enable cell survival, proliferation, differentiation, high transparency, a specific amount of mechanical force, minimally invasive implantation, and other qualities in order to meet the goal of corneal stroma regeneration. We found 86 relevant publications from 1985 to 2022 on the Pubmed website after searching for the terms “chemical materials” and “cornea”. Of these, 24 were about applied research on the cornea. There is a wide variety of chemical materials applied to the cornea, which, in summary, are probably the following.

### 2.1. Chitosan

Although the polysaccharide chitosan has a long history dating back to the 19th century, it has only recently gained popularity in the biomedical and drug delivery industries. Research on chitosan has mainly focused on the field of drug delivery [30,31,32]. Chitosan is a key component in medication delivery systems and has been demonstrated to have antibacterial and wound-healing properties [33]. Unfortunately, histological analysis and assessment of the rabbit cornea’s tensile strength after exposure to chitosan (1% solution) did not reveal enhanced corneal wound-healing [33]. However, in 2018, Xu and his colleagues developed a combination of gelatin, hyaluronic acid, and carboxymethyl chitosan as the carrier for ocular epithelial cells for wound healing. A water-soluble variant of chitosan called carboxymethyl chitosan possesses wound-healing capabilities. During the healing process, gelatin promotes cell adhesion and proliferation. The outcome demonstrated that several corneal epithelial cells were joined together. The cells’ shape and adhesive activity remained normal. The authors noted a full recovery of a rabbit eye region with corneal injury brought on by alkali at the conclusion of the trial [34]. In addition, Felt et al. discovered that chitosan increases the drug’s efficacy in the treatment of dry eye and keratoconjunctivitis sicca(KCS) by extending the drug’s residence duration on the corneal surface. As a result, the scientists suggested using chitosan as a formulation for artificial tears [35]. In terms of safety, scientists found that chitosan is easily biodegraded and removed as early as 1993 [36]. We have also summarized the advantages and disadvantages of the main polymers used in recent years for regeneration (Table 1).

### 2.2. Hydrophilic Gels

Hydrophilic gels have been a research hotspot in recent decades. Three-dimensional (3D) hydrophilic polymer networks in hydrogels allow them to capture large volumes of water or biological fluids while remaining impermeable. Different hydrophilic functional groups (-CONH_2_, -OH, -SO_3_, -CONH, and -COOH),that are connected to the polymer backbone in the polymer chain, as well as the existence of physical and/or chemical crosslinks, account for this feature [60]. The unique class of materials known as hydrogels mimics the three-dimensional tissue environment by seeming to be neither fully liquid nor fully solid [61]. Although there is limited physical connection between polymer networks, the hydrogel does withstand permanent deformation thanks to the creation of strong chemical bonds within it [62]. Two crucial factors for the bioprinting of biomaterials and organs are the bioink and the 3D printing method employed. Hydrogels are excellent bio-ink options. Hydrogels are similar to the natural extracellular matrix and can provide good survival conditions for cell proliferation. Natural hydrogels with strong biocompatibility, biodegradability, swelling, and cytocompatibility include chitosan [40], collagen [63], and gelatin [64]; as a result, they have been a popular subject of study in bioink research.

### 2.3. Collagen

Collagen is partially hydrolyzed to produce gelatin, a biodegradable polypeptide. Connective tissues, including cartilage, corneas, tendons, ligaments, blood vessels, and dentin, are kept intact by collagen. The structural proteins in the extracellular matrix (ECM) of many tissues make up the natural protein collagen, which is found in the human body. Different animal tissues include collagens in a variety of structural and hierarchical arrangements. While type II and IV collagen are generally found in cartilage and basement membranes [41], type I collagen is more prevalent in bone tissue, skin, tendons, ligaments, and the cornea [65]. The center regions of the molecules and the determinant polypeptide structures in the three spiral chains are what cause collagen’s low antigenicity. Its use in biological domains has been severely constrained by this defect [43]. An example is the use of a type I collagen–gelatin hydrogel for ocular tissue engineering by Goodarzi et al. [66] in 2019. Human bone marrow mesenchymal stem cells (hBM-MSCs) were employed as cell evaluations in this investigation, along with N-(3-dimethylaminopropyl)-N_0_-ethylcarbodiimide hydrochloride (EDC) and N-hydroxysuccinimide (NHS). This study looked at how the amount of collagen affected the various characteristics of the gelatin hydrogel. For instance, it was discovered that the collagen–gelatin had an amazing porosity structure and that the composite hydrogel had superior transparency than gelatin-based hydrogels, both of which improved cell adhesion. Despite their suitability for bioprinting, collagen-based materials have some drawbacks, such as a liquid state at low temperatures, the creation of a fibrous structure with rising temperature, or neutral pH.

In the cornea, type I collagen predominates, with lesser levels of the other collagen proteins. Collagen has therefore been a widely used substance for creating scaffold materials that resemble native corneal components. Orwin and Hubel [67] explain the formation of a collagen sponge that fosters the development of endothelial, stromal, and corneal epithelial cells. In Wu’s work [68], collagen was added to a gelatin/alginate combination and effectively printed utilizing the extrusion bioprinting technique. After printing, human corneal epithelial cells (HCECs) may attain a high cellular viability of 94.6% thanks to this matrix’s improved capacity to replicate tissue-specific ECM.

Sodium alginate and type I collagen were employed as a bioink by Isaacson et al. to extrusion-print corneas [69]. The native human corneal stroma was printed by scientists using the appropriate support structures. The corneal keratocytes could be contained by the low-viscosity bioink for a period of seven days. Additionally, various collagen concentrations were employed to identify the corneal tissue’s ideal viscosity and tolerable stiffness. Additionally, the collagen content in the study was altered to enhance the printability of the bioink. Alginate, on the other hand, enhanced the transparency qualities. Another study employed collagen-based hydrogels as a bioink to produce corneal stromal [70]. This bio-printed structure’s optical characteristics were comparable to those of a natural corneal stroma, and the hydrogel’s mechanical qualities made it well suited for the engineering of corneal tissue. According to Kutlehria et al.’s [42] bio-printed’s structure for corneal stromal equivalents, calcium chloride was employed to crosslink the gelatin–collagen bioink. The scaffolds in this study were printed using extrusion and stereolithography techniques. Human corneal keratocyte cells were used in the bioink, like in previous experiments [45,70], to print a cell-filled structure.

### 2.4. Gelatin Methacrylate

GelMA is a potentially useful bioink for biofabrication applications that is inexpensive, simple to synthesize, and biocompatible. Additionally, when GelMA and the photo-initiator are combined and subjected to UV light, fast crosslinking can occur both during and after extrusion [71]. Due to its cell-adhesive RGD (arginine, glycine, and aspartate) patterns and MMP-degradable amino acid chain, GelMA also possesses the necessary biological characteristics. As a result, it can encourage many cell types’ adhesion, spreading, and proliferation [44]. GelMA materials have been used in corneal tissue engineering [72]. When a subconjunctival injection of this printed hydrogel to encapsulate conjunctival stem cells was performed in 2020 by Zhong and colleagues [73], it was discovered that the printed hydrogel was able to sustain the vitality and proliferative potential of the stem cells. Bektas [74] used the GelMA hydrogel as a bioink to 3D bioprint the corneal stroma and simulate the keratocytes growth environment and found that cell viability in this printed crystalloid maintained its original activity with only 8% weight loss after three weeks and good transparency.

### 2.5. Silk Fibroin

A vital protein that comes from the silkworm is called silk fibroin. Due to its exceptional transparency, remarkable biocompatibility, capacity to encapsulate cells, and lack of cytotoxicity, Kim et al. [45] employed a cornea-derived decellularized extracellular matrix as a bioink in a different work from 2019. In this investigation, the bioink nearly completely preserved the viability of the cells (human turbinate mesenchymal stem cells). The hydrogels were discovered to be biocompatible with rabbit and mouse cornea using one- and two-month in vivo experiments. Silk fibroin, a novel type of bioink with exceptional mechanical characteristics and biocompatibility for tissue engineering, was created by Gong et al. via a two-step process. It was possible to create a double-network hydrogel for 3D printing that exhibits acceptable shear-thinning characteristics and has high strength and strong resilience [47].

### 2.6. Glycerol Sebacate

Due to the hydroxyl groups’ ability to cross-link and create hydrogen bonds with one another, poly (glycerol sebacate) (PGS) is a flexible elastomer with a nonlinear stress–strain behavior that can almost entirely recover from significant deformations [49]. For the first time, Wang reported the synthesis of PGS as a robust biodegradable polyester [48]. Salehi et al. developed a technique for producing semi-translucent aligned nanofibrous PGS-PCL, a co-blended scaffold of nanofiber PGS-PCL structured using a modified electrospinning approach [75]. HCEC oriented in the direction of the fibers on unidirectional PGS-PCL mix nanofibrous scaffolds, and the presence of PGS within blended scaffolds (4:1) boosted cell survivability and proliferation in comparison to pure PCL scaffolds [76]. Polycaprolactone (PCL) is a semi-crystalline, hydrophobic, thermoplastic polyester that has received FDA approval for use in humans. The electrospinning of random and aligned nanofiber matrices was done using polycaprolactone (PCL) as the fundamental polymer [77]. Human corneal keratocytes (HCK) and HCEC were employed by Piotr Stafiej et al. They were cultivated on a combination of PCL and PCG, and they discovered that both HCEC and HCK displayed growth activity [78].

### 2.7. Decellularized Extracellular Matrix

Decellularizing mammalian tissues results in the creation of a decellularized extracellular matrix (dECM), a mixture of naturally occurring polymers. Decellularization maintains the initial tissues’ shape and structure, resulting in tissue-specific microenvironments that preserve cell-specific functions. Methods for decellularization might be electrical, chemical, biological, or a mix of these [79]. Studies revealed that the dECM-based bioink induces the expression of genes in hBM-MSCs [80]. In order to create cellular substrates, Kim et al. [45] employed a cornea-derived decellularized extracellular matrix (Co-dECM) as the raw material for a bioink. The created Co-dECM bioink displayed comparable quantitative measures of collagen and GAG in comparison to genuine corneas and also had a suitable level of visual transparency. Only the Co-dECM group showed the ability of human-nasal-turbinate-derived mesenchymal stem cells (hTMSCs) to differentiate into the keratinocyte lineage. Additionally, by xenografting Co-dECM gels into mice and rabbits for two months and one month, respectively, the produced bioink was much more biocompatible and had no deleterious impact on encapsulated cells utilized for three-dimensional growth. Wang et al. [29] produced a decellularized porcine-corneal-derived hydrogel (DPC) for regenerating the epithelium and stroma in focal corneal defects. When injected into the rabbit cornea, the hydrogel rapidly covered the defect surface, with the complete regeneration of corneal epithelium within 3 days and the recovery of corneal thickness 12 weeks after surgery without significant inflammation or scar formation. Notably, the hydrogel had no deleterious effects on the resident stroma or endothelium.

The characteristics of the materials listed above, along with the findings of previous studies, lead us to assume that GelMA is a better scaffold for corneal stromal regeneration.

## 3. Research into GelMA Material in Corneal Stromal Regeneration

### 3.1. Introduction of GelMA and Cornea Stroma

Cell-based and polymer-based applications in the cornea (Table 2 and Table 3). We would like to further discuss GelMA. Cross-linked networks of hydrophilic polymers that may swell considerably in water are known as hydrogels. The content, polymerization process, and crosslinking density of hydrogels all have a significant impact on their physical and biological characteristics. 

GelMA is a hydrophilic gelatin derivative derived from the original gelatin [87]. GelMA hydrogels have been extensively employed for a variety of biomedical applications, due to its excellent biological qualities and adaptable physical characteristics [88]. The corneal stroma serves numerous important functions within the eye. As the primary refracting lens, it must combine the almost flawless transmission of visible light with accurate form in order to concentrate incoming light. Furthermore, it must be exceedingly durable mechanically in order to preserve the inside contents of the eye [89]. The basic principles of corneal structure and transparency have been known for a long time, and yet X-ray scattering and other methods have recently confirmed that the facts of this structure are far more complex than initially thought, and that the complexity of the structure of the collagenous fibrils provides the shape and mechanical properties of the tissue. Corneal collagen fibrils, being the major load-bearing elements of the lamellae, must resist tensile pressures caused by intraocular pressure and defend the inner ocular tissues from external trauma while staying narrow to enable tissue transparency [89,90,91].

GelMA, as a transparent material, fully meets the requirements of the corneal stroma for transparency, and GelMA can satisfy the biocompatibility and mechanical strength requirements for producing biomaterials, as opposed to other biomaterials made from hydrogels. GelMA offers a good habitat for many types of cells to live in, since it is biocompatible [92], chemically adaptable [93], and biodegradable [88]. According to earlier studies, when cells are grown in three-dimensionally organized hydrogels, they have the ability to modify their surroundings to facilitate migration and spread [94]. GelMA possesses the qualities necessary for a substrate that is perfect for tissue-engineering scaffolds [95].

### 3.2. GelMA Biological Properties

GelMA materials have good mechanical and optical transmittance and are also biocompatible. Mahdavi et al. [96] discovered that scaffolds with 12.5% GelMA concentration were more effective when used to 3D print dome-shaped constructs of human corneal stroma. Compared to 7.5%, GelMA’s mechanical and optical transmittance were superior, and its water content was higher. The water concentration was likewise comparable to that of stromal tissue seen in genuine corneas. The impact of concentration on the cytocompatibility test was a key finding of this investigation. By raising the concentration of GelMA, cytocompatibility was greatly improved.

Chen et al. [97] did an interesting study in which they used 5% GelMA as a lamellar keratoplasty (LKP) graft donor and found that inflammatory cells in the corneas of rats transplanted with GelMA material were significantly reduced; corneal thickness was near normal, and corneal stromal α-SMA and TGFβ1 expression was significantly enhanced. GelMA helped to alleviate corneal stromal fibrosis and reduce the loss of strength due to fibrosis.

Farasatkia et al. [98] developed micro-patterned nano-hybrid films based on SNF(silk nanofibril) and GelMA by the micro-injection molding method. Robust and double-layer micro-patterned bioadhesives not only mimic the structure of normal corneal stromal tissue but also controls the orientation of the cells and prevents the differentiation of these cells into myofibroblasts. The differentiation of these cells into myofibroblasts was prevented. In addition, the step length of micropatterns is comparable to that of cells and can regulate the behavior of cell populations. On the other hand, Farasatkia [99] also discovered that varying the quantities of SNF and GelMA led to various mechanical characteristics and biocompatibility, with the greatest results obtained when the GelMA ratio was 30%. Surprisingly, the optimized SNF/GelMA volume ratio of 30/70 demonstrated more than 85% light transmittance, adequate wettability, tensile strength, E-modulus, elongation, toughness, and proper degradation rate in two distinct conditions after 10 days without inducing mineralization [99]. In addition Xin et al. [100] verified the good hemolysis, cell viability, and proliferation ability of GelMA tubes by using small molecules and human red blood cells(Figure 1). Significant GelMA concentrations produced high mechanical strength, but at the expense of porosity, degradability, and three-dimensional (3D) cell attachment [101]. Scanning electron microscopy (SEM) can show the pore morphology of GelMA. Lee et al. [102] used SEM to scan GelMA and found that the pore diameter of GelMA material was about 5 μm.

Because of its complex structure and keratocyte–fibroblast transition, corneal stroma regeneration has always been difficult. In a study, Kong and his team [100] created a grid poly(-caprolactone)-poly (ethylene glycol) microfibrous scaffold and infused it with GelMA to create a 3D fiber hydrogel construct; the fiber spacing was adjusted to create an optimal construct that simulated the stromal structure and had properties that were most similar to the native cornea. The effects of topological structure (3D fiber hydrogel, 3D GelMA hydrogel, and 2D culture dish) and chemical factors (serum, ascorbic acid, insulin, and -FGF) on the differentiation of limbal stromal stem cells to keratocytes or fibroblasts and phenotype maintenance and in vitro and in vivo tissue regeneration were investigated. The findings showed that fiber hydrogel and serum-free media work together to offer an ideal environment for keratocyte phenotyping and corneal stroma regeneration.

In conclusion, GelMA materials are produced by chemically modifying the natural polymer gelatin. By changing the degree of methacrylate substitution, GelMA prepolymer concentration, initiator concentration, and UV irradiation time, several physical characteristics of GelMA (such as mechanical properties, pore size, degradation rate, and swelling ratio) may be readily altered. In addition, the inclusion of cell adhesion RGD motifs and MMP degradable amino acid sequences ensures that the resultant GelMA maintain the outstanding biocompatibility and bioactivity of gelatin. GelMA may be used to create 3D cell-loaded structures that imitate the structure of real tissues, opening up new possibilities for tissue engineering and regenerative medicine. Another major field of research is the development of hybrid hydrogels by combining GelMA with other materials, such as inorganic particles, carbon compounds, biopolymers, and synthetic polymers. This method enables the creation of hybrid materials that combine the favorable qualities of other components (for example, mechanical capabilities and electrical conductivity) with GelMA’s bioactivity. More innovative materials and applications based on GelMA materials will be developed as a result of their outstanding biological characteristics.

## 4. Application Perspectives of GelMA Materials in Corneal Stroma Regeneration

GelMA are obtained by reacting methacrylic anhydride with gelatin and optically cross-linked under a photoinitiator. They are chemically tunable, biocompatible, low immunogenic, biodegradable, highly transparent, can promote cell adhesion and proliferation, and can meet the biofunctional and mechanically tunable requirements of most tissue-engineering applications by customizing the synthesis process or adding biomaterials(Table 4). For example, 7% GelMA has higher equilibrium water content and porosity, better optical properties, and hydrophilicity, which are more favorable for the growth and proliferation of BMMSCs, while 30% GelMA has the best mechanical properties and is more favorable for promoting the differentiation of BMMSCs to keratocytes. Some investigators found that the modification of GelMA could promote nerve regeneration. In our preliminary pre-experiments, we also found that GelMA are highly transparent, injectable, and easy to shape, making them an ideal scaffold system for corneal stromal regeneration. However, GelMA concentration, photoinitiator concentration, photopolymerization method, curing time, UV dose, cross-linking conditions, and geometric characteristics still need to be explored to prepare a scaffold system suitable for the proliferation and differentiation of LNCs and the degradation time to meet the demand for the effective regeneration of corneal stroma.

We have successfully isolated keratocytes from corneal tissue and proliferated keratocytes in GelMA (Figure 2a). We found that keratocytes can be grown stably in GelMA material (Figure 2c), and the GelMA material in which keratocytes are grown has good light transmission and transparency (Figure 2b). In ophthalmology, corneal diseases, such as keratitis and corneal epithelial cell loss due to trauma, often result in a complete loss of transparency and function of the cornea, which ultimately ends in blindness. But corneal donors are again extremely scarce, and according to statistics, for every 70 patients worldwide who need a corneal transplantation, only one is able to obtain a corneal donor [12]. Therefore, scientists have been working to find an alternative to corneal donors, and keratocytes can grow stably in GelMA material with good transparency and stability. Therefore, we believe that the GelMA material is somehow capable of promoting the growth of keartocytes in vitro, and as the cells grow and produce ECM, the GelMA material gradually degrades and eventually all of which is replaced by corneal stromal, which is expected to bring a fundamental solution to the shortage of corneal transplant donors (Figure 2).

## 5. GelMA in the Future

In conclusion, GelMA is a 3D cell-loaded structure that can be used to create structures that mimic real tissues, opening up new avenues for tissue engineering and regenerative medicine. GelMA is chemically tunable, biocompatible, low immunogenic, biodegradable, highly transparent, and promotes cell adhesion and proliferation, making it suitable for most tissue engineering applications. Most tissue engineering applications' biofunctional and mechanical tunability requirements can be met by customizing the synthesis process or adding biomaterials. Because GelMA is highly transparent, injectable, and easy to mold, it is an ideal scaffold system for corneal stromal regeneration, according to our findings. As a result, GelMA holds great promise for future applications in corneal regeneration and is expected to fundamentally address the corneal donor shortage.

## Figures and Tables

**Figure 1 polymers-14-04227-f001:**
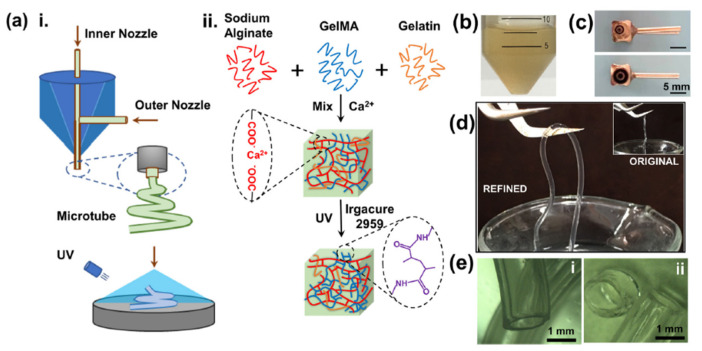
Preparation principle and basic properties of GelMA/gelatin/SA composite hydrogel: (**a**) The GelMA/gelatin/SA composite hydrogel is double-crosslinked using a calcium chloride solution and UV irradiation; (**b**) Preparation of the composite hydrogel precursor solution; (**c**) Different sizes of coaxial nozzles for 3D-bioprinting (two layers: inner diameter = 3.5 mm, outer diameter = 1.37 mm; three layers: inner diameter = 0.61 mm, middle layer = 1.25 mm, outer diameter = 2.27 mm); (**d**) 3D-printed artificial blood vessels optimized by parameters. The upper inset image shows the vessel before optimization and a (**e**) 3D video inspection microscope observation of artificial blood vessels. (Provided by Prof. Yin, Smart Materials in Medicine, 2022, 3: 199–208).

**Figure 2 polymers-14-04227-f002:**
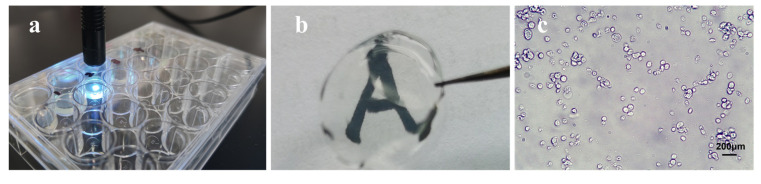
(**a**) GelMA material and 1% elicitor were exposed to UV light for 50 s; (**b**) GelMA solidified after UV irradiation has good transparency; (**c**) Keratocytes were grown in GelMA material.

**Table 1 polymers-14-04227-t001:** Polymer application in regeneration.

Name	Natural/Sythetic	Advantage	Disadvantage	Form
Chitosan	Natural [37]	Antibacterial and easily biodegraded and removed [33,36,38]	Not revealing enhanced corneal wound-healing [33]	Natural biopolymer [37]
Hydrophilic gels	Natural/sythetic [39]	Strong biocompatibility, biodegradability, swelling, and cytocompatibility [40]	Some toxicity reported [39]	Three-dimensional(3D) hydrophilic polymer
Collagen	Natural [41]	Close to natural corneal stroma [42]	Low antigenicity and the liquid state at low temperatures [43]	As bioink for 3D printing [42]
Gelatin methacrylate (GelMA)	Synthetic [44]	Adhesion, spreading, and proliferation of many cells [44]	There are no studies in the corneal stroma.	As bioink for 3D printing
Silk fibroin	Natural [45]	Excellent transparency, biocompatibility, and low cytotoxicity [45].	Poor mechanical performance [46]	A novel type of bioink [47]
Poly(glycerol sebacate) (PGS)	Synthesis [48]	Flexible elasticity with extremely nonlinear stress–strain behavior and biodegradability [48,49].	Harsh synthesis conditions, rapid degradation rates, and low stiffness [50]	Synthesis of PGS as a robust biodegradable polyester [48]
Decellularized Extracellular Matrix (dECM)	Synthesis [45]	Excellent biocompatibility for three-dimensional cell growth [45]	Limited to small tissues or organs [51]	As bioink for 3D printing [45]
Extracellular matrix (ECM)	Natural	Good preservation of the natural ECM structure in corneas [52]	The presence of immune rejection [53]	Natural cellular matrix material [53]
Decellularized SMILE scaffolds	Natural/synthesis [54]	MSC development into corneal epithelial cells can be aided by decellularized lenticules [54].	Standard methods are not widely accepted and are only carried out in a few countries [55].	SMILE-derived stromal lenticules
Poly(ester urethane) urea(PEUU)	Synthesis [56,57]	Mimicking the human corneal stromal tissue [56]	-	A highly organized collagen–fibril construct
Electrospinning	Synthesis [58]	Simulating the structure of the ECM in the natural corneal stroma [56]	Electrostatic spinning does not show uniform inter-fiber spacing, resulting in optically opaque [58]	Artificial fiber
Fish scale derived biocornea	Natural [58]	Good mechanical strength and reasonable optical properties [25]	-	Collagen scaffold
Poly(methyl methacrylate) (PMMA)	synthetic	PMMA structures in rabbit eyes were relatively well retained after 24 months [59].	PMMA is only available in combination with titanium for artificial corneas [59]	

**Table 2 polymers-14-04227-t002:** Cell therapy application in the cornea.

Cells	Method	In Vivo/Vitro	Result	Publication Time	Author
Bone marrow mesenchymal stem cells (BMMSC)	Differentiation into corneal epithelial cells can be achieved in 10 days of culture on amniotic membrane	In vitro	CK3 and p63 expression was significantly enhanced	2014	Rohaina et al. [81]
Oral mucosal epithelial cells	In vitro induction of oral mucosal epithelial cells using human oral mucosal fibroblasts (HOMF) as trophoblast cells for the treatment of (Corneal epithelial stem cell deficiency) CESD	In vivo	Oral mucosal epithelial cells can differentiate into corneal epithelial cells to treat corneal limbal stem cell deficiency	2020	O’callaghan et al. [82]
Dental pulp stem cells (DPSC)	Reconstruction of corneal surface by DPSC in the form of amniotic cell sheets in a rabbit model of CESD	In vivo	Clearer corneas and less angiogenesis in rabbits with DPSC group intervention	2017	Kumar et al. [83]
Induced pluripotent stem cells (iPSC)	Induced differentiation using fibroblast-derived induced pluripotent stem cells (iPSC)	In vitro	Can be induced into PAX6(+) and K12(+) corneal epithelial cells after 12 weeks	2017	Hayashi et al. [84]
Adipose stem cells (ASC)	ASC were also found to induce differentiation into corneal epithelial cells in a laser-induced mouse model for the treatment of CESD in mice	In vivo	Significant healing of corneal epithelial wounds in CESD mice	2017	Zeppieri et al. [85]
Limbal niche cell (LNC)	Subconjunctival injection of LNC cells in a model of CESD	In vivo	LNC can effectively promote the healing of corneal epithelium	2020	Li et al. [86]

**Table 3 polymers-14-04227-t003:** Application of various polymers in cornea.

Polymers	Method	Application Object	Result	Publication Time	Author
Chitosan	Chitosan 1% was applied to rabbits with central corneal injury, and the eyes were spotted 3 times daily for 1 week.	Rabbits	No difference	1987	Sall, K N et al. [33]
Carboxymethyl chitosan, hyaluronic acid, and gelatin	Application of mixed preparations of carboxymethyl chitosan, hyaluronic acid, and gelatin to alkali-burned rabbit corneas	Rabbits	Completely healed	2018	Xu et al. [34]
Chitosan	Solutions containing 0.5% *w*/*v* chitosan were assessed for antibacterial efficacy against *E. coli* and *S. aureus* in vitro.	*E. coli* and *S. aureus*	Effective in inhibiting bacterial growth	2000	Felt O et al. [35]
GelMA	The subconjunctival injection of this printed hydrogel encapsulates conjunctival stem cells.	Rabbits	Sustain the vitality and proliferative potential of the stem cells	2020	Zhong et al. [73]
Collagen and gelatin	For the engineering of corneal tissue, type-I collagen–gelatin hydrogel	In vitro	In addition to increasing Young’s modulus and stiffness, COL-I also boosts optical characteristics.	2019	Goodarzi, H. et al. [66]
Silk fibroin	Co-dECM as a bioink; the bio-ink was injected into mice and rabbits.	Mice and rabbits	Similar to clinical-grade collagen	2019	Kim, H. et al. [45]
Poly (glycerol sebacate) (PGS)	PGS-PCL nanofibrous scaffolds using a modified electrospinning technique, culture HCEC in PGSPCL nanofibrous scaffolds	In vitro	Cell proliferation and viability compared to pure PCL scaffolds was improved in the presence of PGS within blended scaffolds (4:1 specifically).	2017	Salehi et al. [76]
Decellularized porcine-cornea-derived hydrogels	An injectable and transparent hydrogel for the regeneration of epithelium and stroma in localized corneal lesions was developed.	Rabbit	The rabbit corneal epithelium regenerated in 3 days, and corneal thickness returned to normal 12 weeks after surgery without significant inflammation or scar formation.	2020	Wang et al. [29]
Polycaprolactone (PCL)	HCEC and human corneal keratocytes (HCK) cultured in PCL-based matrices	In vitro	HCEC and HCK growth was observed on all aligned PCL-based matrices	2016	Stafiej, Piotr et al. [78]
Decellularized extracellular matrix (dECM)	Co-dECM as a bioink for corneal regeneration; the Co-dECM gel was heterologously implanted into mice and rabbits for two months and one month.	Mice and rabbits	Only the Co-dECM group showed the ability of hTMSCs to differentiate into a keratocyte lineage.	2019	Kim, H. et al. [45]
Collagen	Collagen was successfully printed using extrusion bioprinting technology by adding it to a gelatin/alginate system.	In vitro	HCECs can achieve a high cellular viability of 94.6 ± 2.5% after printing.	2016	Wu et al. [47]

**Table 4 polymers-14-04227-t004:** Application of GelMA material in keratocytes in recent years.

Application	Concentration of the GelMA	Result	Publication Time	Author
GelMA as a bioink to 3D bioprint the corneal stroma	-	Only 8% weight loss after 3 weeks with good transparency.	2019	Kilic Bektas, C. et al. [74]
Structure of 3D printed corneas using different concentrations of GelMA material to compare biological properties	12.5% and 7.5%	The mechanical and optical transmittance of 12.5% were superior.	2020	Mahdavi et al. [96]
Taking GelMA as the donor material for rat lamellar corneal transplantation	5%	GelMA effectively improves repair after corneal injury in rats.	2021	Chen et al. [97]
SNF/GelMA hybrid films	0.5 wt%	Significantly improved biocompatibility after mixing with SNF (30/70).	2020	Asal Farasatkia [99]
Induction of keratocytes regeneration in vitro and in vivo using GelMA	5%, 10% and 15%	3D GelMA can induce the regeneration of damaged corneal stroma in vitro and in vivo.	2020	Kong, Bin [103]

## Data Availability

Not applicable.

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
