# Peer review of "Application Prospect and Preliminary Exploration of GelMA in Corneal Stroma Regeneration"

_polymers, 2022, doi:10.3390/polym14194227_

Round 1

Reviewer 1 Report

Dear Editor in Chief,

Dear Editor in Chief,

In a review article entitled "Application Prospect and Preliminary Exploration of GelMa hydrogel in Corneal Stroma regeneration," the authors aimed to present the current application of GelMA in Corneal regeneration briefly; however, the topic is not novel enough, and the review is not well designed enough to consider for publication. Follows are my suggestions for improving this review article for considering for publication in your prestigious journal:

1. Since the topic is not novel enough, I have suggested that authors turn it into a more comprehensive review, including all polymers for regeneration of corneal stroma, and omit the word hydrogel and chemical material in the manuscript. The authors are also able to classify the polymers into natural and synthetic polymers and include their structure in the manuscript. They are able to discuss the advantages and disadvantages of each polymer structure for corneal stroma regeneration in more detail. For each polymer, they are able to add different compositions and scaffolds with those polymers for corneal stroma regeneration and mention the structure type which is built by that polymer and mention if they are in hydrogel form, nanofiber form, or 3D printed form.

2. This review does not have enough images to represent the application of this hydrogel and other hydrogels which are commonly used for regeneration and treatment of corneal stroma regeneration. I recommend adding graphical abstract. Moreover, the authors should ask for the permission of original papers to use their in vivo results’ images besides discussing their results in this review article.

3. In the introduction, the authors mentioned the cell therapy application. It will be useful for readers to know about the studies which used the different polymers alone or with cells and their outcomes in a separate table.

Author Response

Reviewer #1 (Comments for the Author (Required)): In a review article entitled "Application Prospect and Preliminary Exploration of GelMa hydrogel in Corneal Stroma regeneration," the authors aimed to present the current application of GelMA in Corneal regeneration briefly; however, the topic is not novel enough, and the review is not well designed enough to consider for publication. Follows are my suggestions for improving this review article for considering for publication in your prestigious journal:

Point 1: Since the topic is not novel enough, I have suggested that authors turn it into a more comprehensive review, including all polymers for regeneration of corneal stroma, and omit the word hydrogel and chemical material in the manuscript. The authors are also able to classify the polymers into natural and synthetic polymers and include their structure in the manuscript. They are able to discuss the advantages and disadvantages of each polymer structure for corneal stroma regeneration in more detail. For each polymer, they are able to add different compositions and scaffolds with those polymers for corneal stroma regeneration and mention the structure type which is built by that polymer and mention if they are in hydrogel form, nanofiber form, or 3D printed form.

Response 1: Thank you for your suggestion, I strongly agree with your comments, the topic of this manuscript is not novel, but this review focuses on the role of Gelma material in the regeneration of corneal stroma. Before writing this manuscript, we searched the Pubmed database with the keywords "corneal stroma", "Gelma" and "regeneration", and we did not find any relevant reviews. Therefore, we think that we should write such an overview. And natural and synthetic polymers in corneal stromal regeneration has been reviewed in some other researches.

We have further tabulated the polymers mentioned in the text, detailing the advantages, disadvantages and forms of each polymer (Please see table 1).

Point 2: This review does not have enough images to represent the application of this hydrogel and other hydrogels which are commonly used for regeneration and treatment of corneal stroma regeneration. I recommend adding graphical abstract. Moreover, the authors should ask for the permission of original papers to use their in vivo results’ images besides discussing their results in this review article.

Response 2: Thank you for your suggestion. Our focus in this manuscript is on Gelma material, and we have cited a picture that describes how Gelma material is synthesized and describes Gelma material in vascular regeneration (Please see Figure 1). Our citation was approved by the authors of the original paper

Point 3: In the introduction, the authors mentioned the cell therapy application. It will be useful for readers to know about the studies which used the different polymers alone or with cells and their outcomes in a separate table.

Response 3: Thank you very much, we have reviewed the studies related to the relevant cells for corneal stromal regeneration and have summarized the applications of corneal cell regeneration, please consult Table 4, which in combination with Table 3 is polymers, basically describes the current research progress of polymers and cell therapy.

Reviewer 2 Report

The article is well written and provides significant insight into the use of GelMa as a potential material for corneal stroma regeneration. I do have the following comments/suggestions that I think might add value:

1. In Line 154, replace carbodamide with carbodiimide.

2. In Line 183, provide references for "previous experiments".

3. Include more confocal microscopy images of keratocyte laden GelMa scaffolds indicating cell viability and proliferation.

4. What are the mechanical properties of the corneal stroma tissue? And how well does GelMa based scaffolds compare?Are the GelMa scaffolds able to match the mechanical properties of the native stromal tissue? Include references from literature to support this.

5. Include SEM micrographs of GelMa scaffolds from literature to show their porosity as porosity plays an important role in cellular proliferation.

6. Figure 1 and its caption should be aligned.

7. Thorough grammar check and proof reading are required.

Author Response

Reviewer #2 (Comments for the Author (Required)): The article is well written and provides significant insight into the use of GelMa as a potential material for corneal stroma regeneration. I do have the following comments/suggestions that I think might add value:

Point 1: In Line 154, replace carbodamide with carbodiimide.

Response 1: Thank you for your suggestion. We have checked the word “ethylcarbodiimide” ,and already replaced carbodamide with carbodiimide.

Point 2: In Line 183, provide references for "previous experiments".

Response 2: Thank you for your suggestion. We have add the references that for the previous experiments.(please see line 185 in red)

Point 3: Include more confocal microscopy images of keratocyte laden GelMa scaffolds indicating cell viability and proliferation.

Response 3: This is a very good suggestion, I tried to search "confocal and GelMa" in the PUBMED database, but the search result is "0", unfortunately I didn't find relevant confocal images of Gelma, but I mentioned Gelma in the article that researcher tried to culture human red blood cells on Gelma material and found that Gelma material has good viability and proliferation. Our future research will incorporate confocal microscopy images of keratinocytes loaded with GelMa scaffolds to show cell viability and proliferation. (Line 310-313 in red).

Point 4: What are the mechanical properties of the corneal stroma tissue? And how well does GelMa based scaffolds compare?Are the GelMa scaffolds able to match the mechanical properties of the native stromal tissue? Include references from literature to support this.

Response 4: Thank you for your suggestion. We have described the corneal stromal tissue and compared it with Gelma in terms of mechanical properties and transparency, illustrating the superiority and possibilities of Gelma in becoming a corneal stromal material (line 263-275 in red)

Point 5: Include SEM micrographs of GelMa scaffolds from literature to show their porosity as porosity plays an important role in cellular proliferation.

Response 5: Thank you very much for your suggestion, I'll check the relevant literature, there is only one research in 2019 about the SEM micrographs of Gelma,and found that the pore diameter of Gelma material was about 5um (line 313-316 in red)

Point 6: Figure 1 and its caption should be aligned.

Response 6:  We really appreciate your suggestions. We have already adapted Figure 1 and modified it accordingly (See Figure 2).

Point 7: Thorough grammar check and proof reading are required.

Response 7: Thank you for your suggestion. We did a grammar check of the manuscript and did our best to make changes.

Round 2

Reviewer 1 Report

The article is acceptable in the present format.